# Rationally designed Fe-cyclopentadienone with unique orientations for efficient asymmetric hydrogenation of acylsilanes

Chaochao Xie[1,4], Bo-Xuan Yao[2,4], Kwok-Chung Law[1], Xumu Zhang [3], Shao-Fei Ni [2] ✉ & Xuefeng Tan [1] ✉

Fe-cyclopentadienone complexes have been widely utilized in various hydrogenation and dehydrogenation catalytic processes, yet their applications have largely been restricted to non-asymmetric versions. This limitation is primarily due to the considerable challenge of constructing an efficient chiral environment around the active iron center. In this study, we present a structurally distinctive chiral Fe-cyclopentadienone complex with excellent enantiocontrol capabilities. This new iron complex features bulky side arms oriented downward toward the cyclopentadienone plane, which create an ideal chiral environment in front of the catalytically active iron center. It demonstrates excellent performance in the catalytic asymmetric hydrogenation of acylsilanes, exhibiting both high reactivity and selectivity. The broad substrate scope, encompassing aryl-, alkenyl-, and alkyl-acylsilanes, along with successful gram-scale synthesis, underscores its potential applications in pharmaceutical synthesis. Experimental and DFT studies reveal the structural stability and rigidity of the catalyst during catalytic intervals. Additionally, weak interactions between the catalyst and the silyl group in the substrate play a critical role in achieving efficient enantioselectivity. More importantly, this type of chiral iron complex also shows excellent catalytic reactivity and selectivity for asymmetric transfer hydrogenation, utilizing *i*-PrOH as the hydrogen source.

Transition metal-catalyzed asymmetric hydrogenation (AH) has been widely explored due to its practicality in industrial production, particularly in pharmaceutical companies[1]. Since the first industrial application of Rh/DIPAMP by Knowles in 1974, and its subsequent recognition with the Nobel Prize in 2001, asymmetric hydrogenation has experienced a golden era of research centered around noble transition metals, such as ruthenium, rhodium, and iridium[2–9]. A significant shift occurred around 2010, when chemists began focusing on earth-abundant transition metals, such as manganese, iron, cobalt, nickel, and copper[10–19]. Among these, iron stands out as a promising candidate for replacing noble transition metals in industrial applications, owing to its status as the second most abundant metal in the Earth's crust and negligible toxicity to humans[20]. Although several sophisticated iron complexes have been developed for asymmetric hydrogenation[21–26], the quest for highly efficient and practical chiral Fe-catalysts, with the ultimate goal of industrial application, continues unabated.

Due to the smaller 3d orbitals and higher electronegativity of iron compared to its congener ruthenium, iron-based catalysts exhibit distinct catalytic behaviors. For instance, iron is more likely to

[1]Department of Chemistry, City University of Hong Kong, Kowloon Tong, Hong Kong, China. [2]College of Chemistry & Chemical Engineering, Shantou University, Shantou, China. [3]Department of Chemistry, the Grubbs Institute, and Medi-X Pingshan, Southern University of Science and Technology, Shenzhen, Guangdong, China. [4]These authors contributed equally: Chaochao Xie, Bo-Xuan Yao. ✉e-mail: sfni@stu.edu.cn; xuefetan@cityu.edu.hk

**Fig. 1 | Overview of the development of Fe-CPDs and our work. a** Introduction to the Catalytic Model of Fe-CPD, categorized as a bifunctional catalyst, along with an explanation of the challenges in constructing an enantioselective Fe-CPD. **b** Rational analysis of the steric requirements for chiral Fe-CPD in the asymmetric hydrogenation of polar double bonds. **c** Our work: a newly developed Fe-CPD features the desired downward extension of the side arms and the steric difference formed, and its performance in asymmetric hydrogenation of acylsilanes. CPD, cyclopentadienone.

undergo single-electron transfer processes and possesses complex valency and spin states[27]. Consequently, chemists have recognized that the classical catalytic modes of ruthenium cannot be directly and simply replicated with iron, which is often perceived as inferior in performance[28]. However, a notable exception is the Fe-cyclopentadienone (Fe-CPD) complex (Fig. 1a), which displays a similar concerted catalytic mode and comparable reactivity to its ruthenium analog, Shvo's catalyst[29]. Although Fe-cyclopentadienone was initially synthesized in the 1950s[30] and further explored by Pearson and Knölker in the 1990s[31–33], its catalytic potential was largely overlooked until 2007, when Guan and co-workers demonstrated. It's high efficiency in the hydrogenation of ketones[34,35]. Fe-CPD offers several advantages, including being inexpensive, low in toxicity, easily accessible, phosphine-free, and air-stable. The catalytic mode of Fe-CPD can be attributed to a bifunctional active intermediate, consisting of a Lewis acidic iron center and a Lewis basic oxygen center (Fig. 1a). Based on this bifunctional catalytic

model, a wide range of catalytic transformations involving hydrogenation and dehydrogenation have been developed[36–43].

Despite the extensive application of Fe-CPD in various non-asymmetric transformations, the development of enantioselective reactions using chiral Fe-CPD has lagged far behind. The primary reason is the lack of an efficient chiral Fe-CPD catalyst, due to the difficulty in designing an effective chiral pocket around the iron active center. Addressing the challenge of designing an efficient enantioselective Fe-CPD, the Wills group noted that the limited enantioselectivity may result from the long distance between the side arms of the cyclopentadienone ring and the substituents of the substrate in the proposed reduction model (Fig. 1a)[44].

Fe-CPD-catalyzed asymmetric transformations primarily rely on two strategies: 1) the introduction of a chiral cocatalyst and 2) the design of a chiral cyclopentadienone (CPD) backbone. The first strategy has been effective in limited instances, as it requires specific recognition between the substrate and the chiral cocatalyst, such as

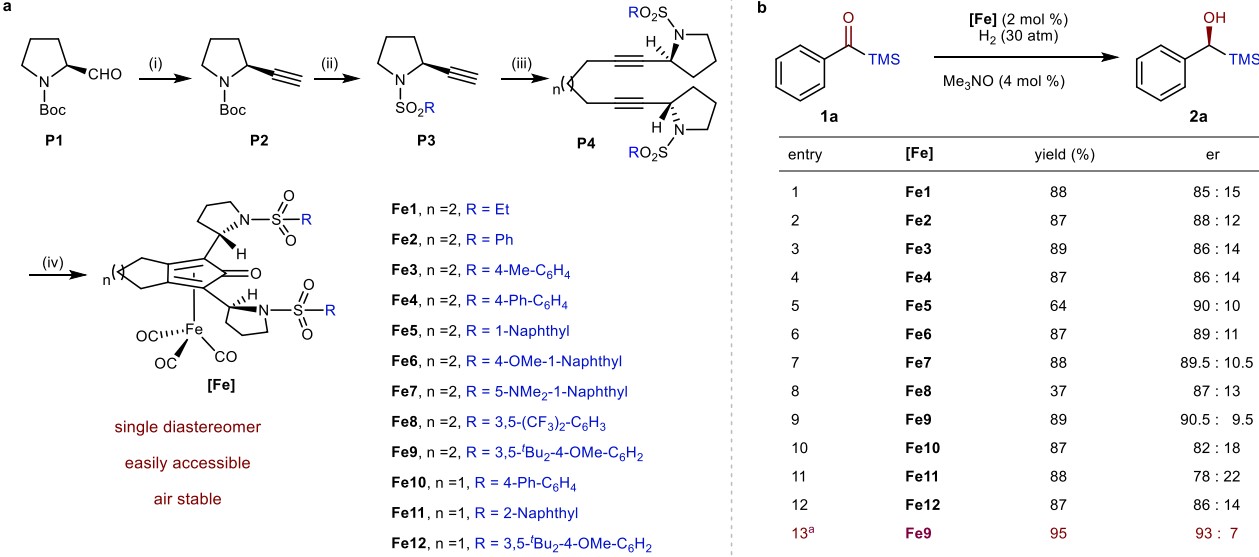

**Fig. 2 | Preparation of chiral Fe-CPDs and the catalytic asymmetric hydrogenation of acylsilane. a** Synthetic route for the Fe-CPDs, reaction conditions: (i) Dimethyl (1-diazo-2-oxopropyl)phosphonate, $K_2CO_3$, MeOH. (ii) TFA, DCM; then $RSO_2Cl$, $NEt_3$. (iii) LDA, 1,3-diiodopropane/1,4-diiodobutane, THF. (iv) $Fe_2(CO)_9$, toluene, reflux. **b** asymmetric hydrogenation of **1a**, reaction conditions: **1a** (0.2 mmol), **[Fe]** (2 mol %), $Me_3NO$ (4 mol %), toluene (0.3 mL), $H_2$ (30 atm), 80 °C, 12 h. [a]*i*-PrOH/$H_2O$ (0.48: 0.12 mL), 60 °C.

imines and chiral phosphoric acid catalysts[45–47]. The second strategy is more appealing due to the direct chiral recognition between the substrate and the chiral Fe-CPD, which is expected to have broader and more general applications. Significant efforts have been devoted to developing chiral CPD backbones, yet no substantial breakthroughs have been achieved, particularly at the practical application level[44,48–51]. To date, the highest enantiomeric excess (e.e.) value achieved is 77%, obtained through the asymmetric hydrogenation of a specific ketone, although this method lacks substrate generality[49].

Given the bifunctional catalytic mode of Fe-CPD, a visionary enantiocontrol model for the asymmetric hydrogenation of polar double bonds, such as ketones and imines, can be established (Fig. 1b, left). The current challenge of designing an efficient enantioselective catalyst can be explained by model **M1**, which features open steric hindrance in front of the catalytically active iron center, making effective enantio-differentiation difficult (Fig. 1b). This conclusion is supported by the crystal structures of catalysts described in earlier reports[44,48–51]. Based on the concerted enantiocontrol model, we are able to figure out a desired and sterically feasible model **M2** and an undesired, sterically crowded model **M3** (Fig. 1b). In this work, we report an chiral Fe-CPD catalyst that simultaneously positions steric side arms downward to the CPD plane and creates two distinct steric environments in front of the catalytically active iron center (illustrated with **Fe4**, Fig. 1c). This type of catalyst demonstrates excellent performance in the asymmetric hydrogenation of acylsilanes, achieving up to a 96:4 enantiomeric ratio (e.r.) and a turnover number (TON) of 870. Moreover, a slight modification of the catalyst enables highly efficient asymmetric transfer hydrogenation, using *i*-PrOH as the hydrogen source.

## Results and discussion
### Catalyst design and test
At the outset of our study, we aimed to incorporate naturally abundant chiral scaffolds, such as amino acids, into the Fe-CPD framework (Fig. 2a). The chiral aldehyde precursor (**P1**), which can be readily synthesized from *L*-proline, was treated with the dimethyl (1-diazo-2-oxopropyl)phosphonate (Bestmann-Ohira reagent) to produce the corresponding alkyne **P2**. Following -Boc removal and the installation of $-SO_2Ar$, **P3** was obtained, which then underwent a nucleophilic reaction with a diiodide to yield the dialkyne **P4**. The final **Fe1–12** complexes were easily synthesized through the coordination of **P4** with $Fe_2(CO)_9$. This synthetic route offers several advantages, including the easy availability of starting materials, high yields, friendly operation, and all products are air-stable. A notable feature is the production of a single diastereomer of the final Fe-complex, which significantly simplifies the isolation process. Starting from the common synthon **P2**, the final complexes **Fe1–12** achieved a global yield ranging from 12% to 46% (see Supplementary Information). Consequently, this type of catalyst holds potential for large-scale production.

Compared to the extensively studied alkyl-aryl ketone substrates, acylsilanes have only been sporadically explored in asymmetric hydrogenation, despite the silyl group's potential for diverse transformations and biological activities. The classical Ru-TsDPEN catalyst has been shown to be effective with aryl acylsilanes but less so with alkyl acylsilanes[52]. Ohkuma et al. successfully applied the Ru-diphosphine-diamine catalytic system for the asymmetric hydrogenation, but limited to the bulkier -TBS substituent, possibly due to the existence of a strong base, which may cause the Brook-rearrangement[53]. We were also keen to apply our newly developed Fe-CPDs for the asymmetric hydrogenation of acylsilanes, addressing the existing unresolved challenges. Under $H_2$ (30 atm) at 80 °C, a preliminary screening of **Fe1–12** revealed that **Fe9** gave the best enantioselectivity (Fig. 2b, entries 1–12). These results indicate that a six-membered ring backbone is superior to a five-membered ring backbone (entry 4 *vs*. 10 and entry 9 *vs*. 12). Moreover, the different types of R substituents on the sulfonyl group do not significantly influence either catalytic reactivity or enantioselectivity. Further optimization, involving a solvent change to $^iPrOH/H_2O$ and a temperature reduction to 60 °C, resulted in the best enantiomeric ratio (e.r.) of 93:7 with high yield (entry 13).

### Substrate scope exploration
With the optimized reaction conditions in hand, we primarily evaluated the substrate scope of Fe-catalyzed asymmetric hydrogenation

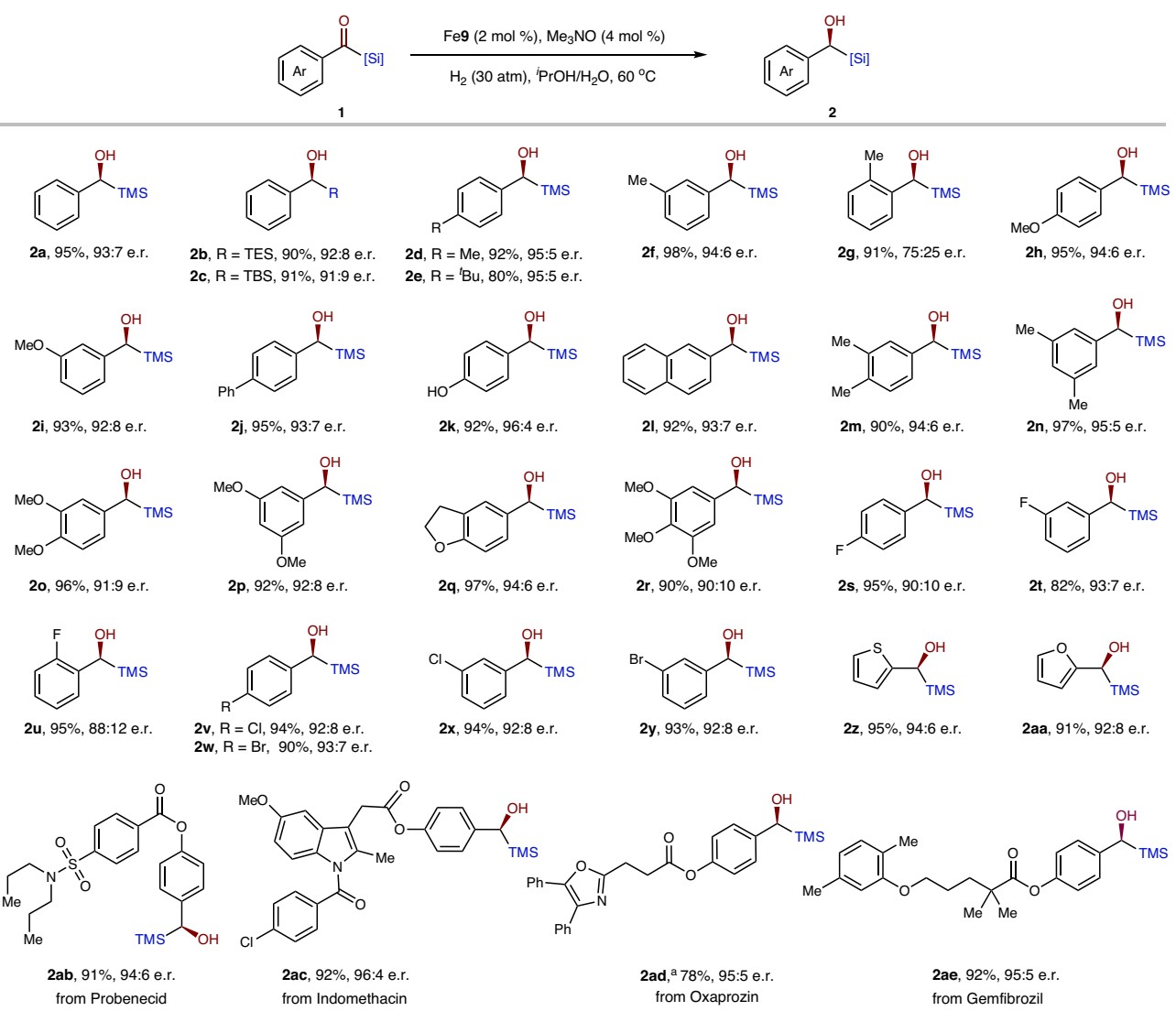

**Fig. 3 | Asymmetric hydrogenation of aryl acylsilanes.** Reaction conditions: **1** (0.5 mmol), **Fe9** (2 mol %), Me₃NO (4 mol %), H₂ (30 atm), $^i$PrOH/H₂O (1.2: 0.3 mL), 60 °C, 12 h. Isolated yields. [a]**Fe9** (4 mol %).

of aryl acylsilanes (Fig. 3). First, changing the silyl groups from -TMS to -TES and -TBS resulted in a slight decrease in the er (**2a**–**c**). Second, while the electronic properties of different substituents did not affect substrate activity, they did have a slight influence on the er, with electron-rich groups generally exhibiting higher enantioselectivity (e.g., **2d**–**e** *vs.* **2v**–**w**). Third, substitutions at the *ortho*-position showed poorer enantiocontrol than those at the *para*- and *meta*-positions (**2d**–**f** *vs.* **2g** and **2s**–**t** *vs.* **2 u**). Additionally, heteroaromatic rings such as thiophenyl and furanyl were well tolerated (**2z** and **2aa**). Notably, some bioactive drugs and natural product-derived acylsilanes were also successfully hydrogenated with high enantioselectivity and reactivity (**2ab**–**ae**). Overall, the aryl acylsilane substrates demonstrated high generality with respect to the catalytic system.

The robustness of this catalytic system was further demonstrated by its successful application to alkyl- and alkenyl-acylsilanes, which have been challenging for the classical Ru-TsDPEN catalyst[52]. With slight modification of the catalyst, specifically **Fe5**, a variety of alkyl- and alkenyl-acylsilanes were efficiently and selectively hydrogenated to the corresponding alcohols (Fig. 4). Notably, aryl- and oxygen-substituted acylsilanes (**4a**–**i**) exhibited higher enantioselectivity compared to bare or fluoro-substituted alkyl acylsilanes (**4k**–**l**). Of particular note, the alkenyl-acylsilanes achieved high enantiomeric ratios without affecting the alkenyl groups (**4m**–**n**).

## Gram-scale synthesis and derivatizations

To further evaluate the robustness of this catalytic system, a turnover number (TON) test was conducted with gram-scales (Fig. 5a). Under a S/C ratio of 1000 at 80 °C, we achieved full conversion with an isolated yield of 87% and a slightly reduced er of 91:9 (Fig. 5a). The low yield is due to the slow decomposition of **2a** in water, which produces benzaldehyde[54]. Lowering the temperature to 70 °C resulted in decreased catalytic reactivity, as an incomplete reaction was observed with S/C = 500. However, a successful gram-scale synthesis was achieved with an S/C ratio of 300 at 70 °C, yielding 86% with an er of 92:8, thereby demonstrating the potential practicality of this catalytic system. Furthermore, the chiral silyl alcohol **2a** can be readily transformed into the corresponding amino product **5a** via a Mitsunobu reaction (Fig. 5b). Another intriguing application is the use of the silyl alcohol as a chiral auxiliary, initially demonstrated by Linderman et al.[55]. The TMS-protected **5b** was able to induce formylcyclohexane (CyCHO) to form an oxocarbenium ion intermediate, which can be enantioselectively attacked by allyltrimethylsilane (Allyl-TMS), yielding **5c** with two chiral centers (Fig. 5b).

## Mechanistic studies

Next, experimental and DFT studies were conducted using **Fe4**, which has been crystallographically characterized, as the standard model to

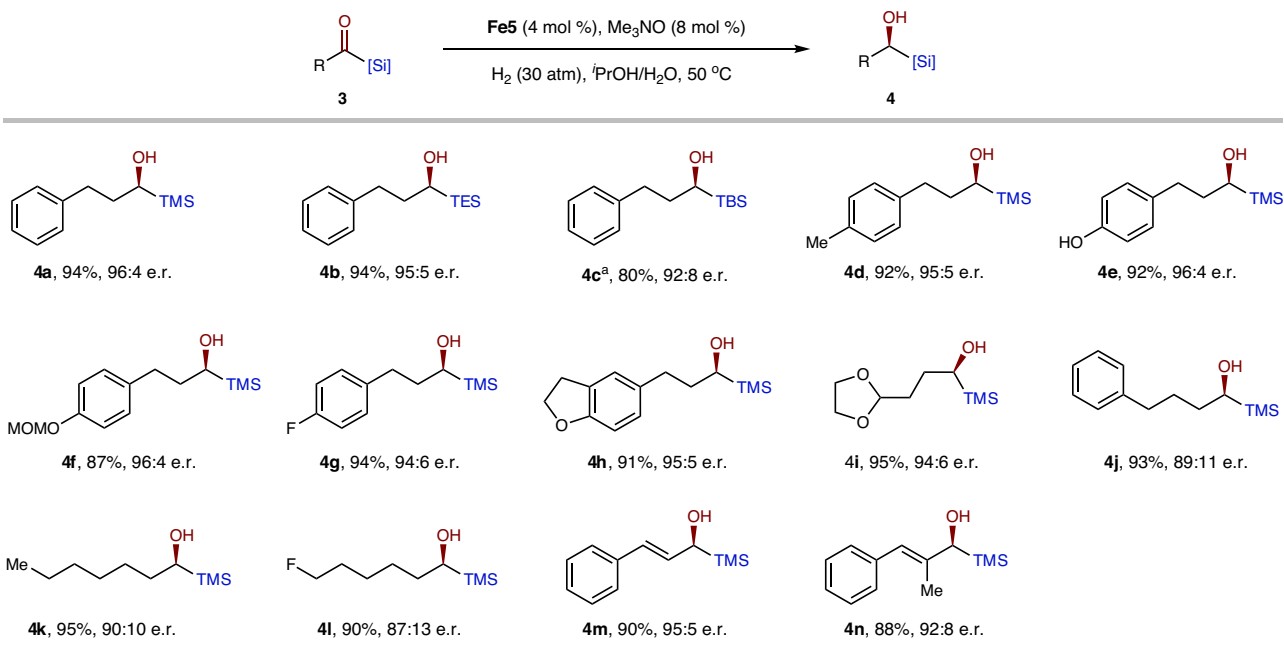

**Fig. 4 | Asymmetric hydrogenation of alkyl and alkenyl acylsilanes.** Reaction conditions: **3** (0.5 mmol), **Fe5** (4 mol %), Me₃NO (8 mol %), H₂ (30 atm), *i*PrOH/H₂O (1.2: 0.3 mL), 50 °C, 12 h. Isolated yields. ᵃ80 °C.

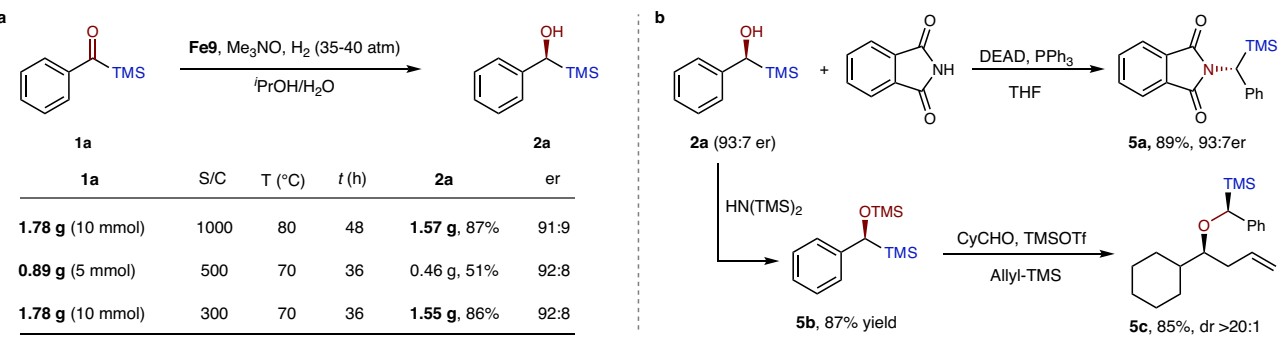

**Fig. 5 | Gram-scale synthesis and derivatizations. a** Turnover number test and gram-scale synthesis. **b** Derivatizations of **2a**. S/C, substrate/catalyst; DEAD, Diethyl azodicarboxylate.

elucidate the catalytic mechanism and enantiocontrol insight (Fig. 6). Initially, when D₂ was used in place of H₂ for the hydrogenation of **1a**, 11% of H atoms were detected at the benzylic position, suggesting possible Fe-H formation, potentially arising from hydrogen transfer from *i*-PrOH (Fig. 6a, entry 1). The transfer hydrogenation process was conclusively confirmed by using *i*-PrOH as the sole hydrogen source, resulting in the successful isolation of the desired product with 10% yield (entry 3). Furthermore, entries 1–2 suggest that the reaction with D₂ proceeds more slowly compared to H₂. A more precise experiment employing H₂/D₂ (1:1) indicated a kinetic difference between H₂ and D₂, with an estimated kinetic isotope effect (KIE) of around 1.5 (entry 4). Combined with DFT calculations, it is proposed that H₂ activation is the rate-determining step (see the Supplementary Information).

The crystal structure of **Fe4** has satisfied all the requirements of the design principle (Fig. 1c, **M2**), with the two ArSO₂– side arms oriented downward relative to the CPD plane, creating a steric difference in front of the catalytic center. However, concerns may arise regarding the potential flexibility of this configuration in solution, which could undermine the previous hypotheses. To investigate the rigidity of the structure, a dimensional NMR test in CDCl₃ was conducted (Fig. 6b). The ¹H–¹H NOESY signal intensities between Hᶜ/Hᵈ–Hᵃ

and Hᵉ/Hᶠ–Hᵇ are positively correlated with the corresponding distances measured in the crystal structure (see the Supplementary Information for details), indicating that the structure in solution closely resembles of the crystal state. Actually, this structural arrangement can be rationalized by comparing the steric hindrance of the –CH₂– and –NSO₂Ar groups adjacent to C3 or C4. Although the physical volume of N–SO₂Ar is larger than CH₂, the –SO₂Ar group can extend peripherally to the Fe center to avoid steric repulsion. In other words, the tetrahedral geometry of –CH₂– is less effective at preventing proximity to the Fe center compared to the pyramidal geometry of –NSO₂Ar. Furthermore, DFT optimized **Fe4** (**Fe4cal**) reveals additional stabilization factors arising from C–H⋯O interactions (b₁ = 2.969 Å, b₂ = 3.399 Å, Fig. 6c) between CO ligands and hydrogens on side arms.

A detailed mechanistic understanding of the reaction was achieved through density functional theory (DFT) calculations (Fig. 6c, Supplementary Data 1). The **Fe4** complex was confirmed to adopt singlet ground states (¹**Fe4cal**), with other spin states exhibiting significantly higher Gibbs free energies (³**Fe4cal** 35.3 kcal·mol⁻¹ and ⁵**Fe4cal** 66.2 kcal·mol⁻¹). First, the pre-catalyst ¹**Fe4cal** is activated with Me₃NO by removal of one CO, followed by a spin crossover via the minimum energy crossing point (**MECP**) to afford the more stable triplet

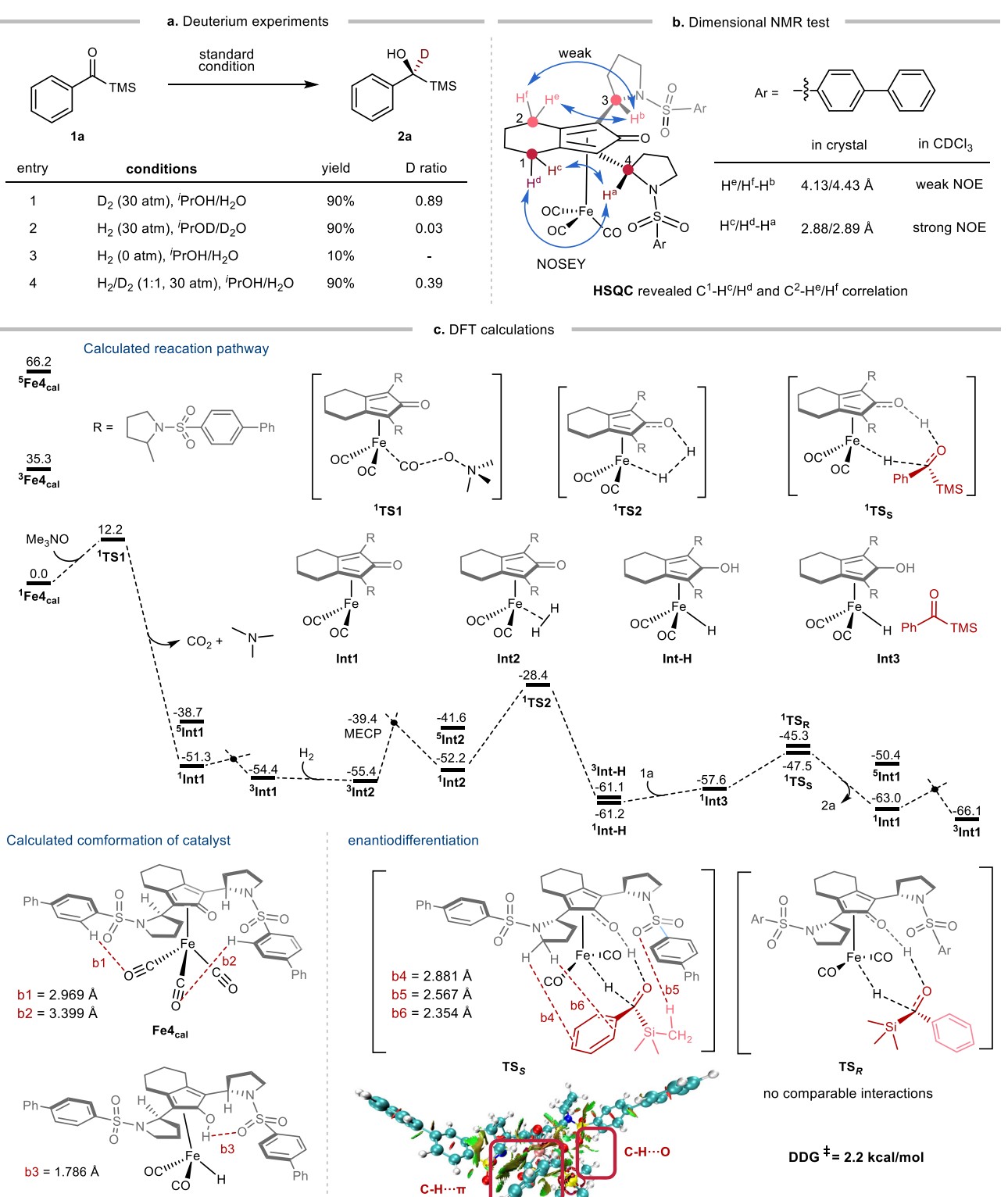

**Fig. 6 | Mechanistic studies based on Fe4. a** Deuterium experiments indicate the presence of transfer hydrogenation and a kinetic isotope effect (KIE) in the catalytic system. **b** Dimensional NMR tests reveal that **Fe4** maintains similar 3D structures in both solid and solution states. **c** DFT calculations of reaction pathway. Calculations identify additional stabilizing factors, such as hydrogen bonds, that help stabilize the 3D structure of **Fe4**. The calculated transition states, **TS$_S$** and **TS$_R$**, show that the C–H···O contact (b$_6$ = 2.354 Å) is the major contributor to enantiocontrol.

intermediate **³Int1**. Then, H$_2$ associates to Fe to form the triplet intermediate **³Int2**, which then undergoes a spin inversion through **MECP** (−39.4 kcal mol⁻¹) to produce the singlet intermediate **¹Int2**. Next, H$_2$ is cleaved to form **¹Int-H**, which step experiences a free energy elevation of 27.0 kcal·mol⁻¹, representing the rate-determining step. Examination of the critical intermediate **¹Int-H** reveals a strong hydrogen bond (b$_3$ = 1.786 Å, Fig. 6c) between O–H···O = S, which helps preserve the chiral pocket. Finally, **1a** is hydrogenated by **¹Int-H** via a concerted transition state (**TS$_S$** = 13.7 kcal·mol⁻¹ and **TS$_R$** = 15.9 kcal·mol⁻¹, Fig. 6c) to yield the desired product **2a**.

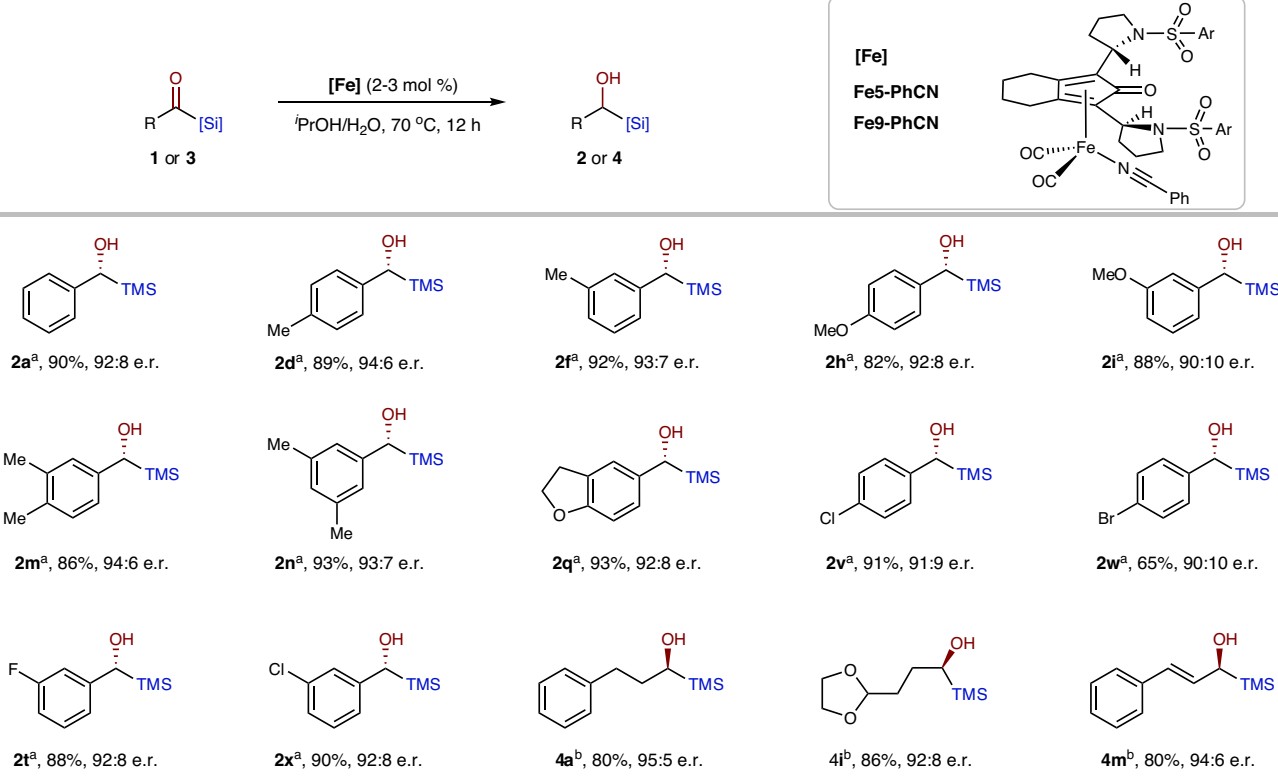

**Fig. 7 | Asymmetric transfer hydrogenation of acylsilanes.** Reaction conditions: **1** or **3** (0.3 mmol), **[Fe]** (2-3 mol %), $^{i}$PrOH/$H_2O$ (1.5: 0.1 mL), 70 °C, 12 h. Isolated yields. [a](−)-**Fe9-PhCN** (2 mol %) was used, (−)-**Fe9** means the enantiomer of the previous **Fe9**. [b]**Fe5-PhCN** (3 mol %) was used.

To gain a deeper understanding of the enantiocontrol, non-covalent interaction (NCI) analysis of the enantiodetermining transition state **TS$_S$** revealed critical stabilizing interactions, including C−H⋯π ($b_4 = 2.881$ Å, $b_5 = 2.567$ Å) and C−H⋯O ($b_6 = 2.354$ Å) contacts (Fig. 6c). While both aryl- and alkyl-acylsilanes are attacked by Fe-H from the same face, we believe that the C−H⋯O contact is more significant in the enantiocontrol transition state[56,57]. In contrast, the disfavored transition state **TS$_R$** lacked comparable interactions, providing a structural basis for the observed enantiocontrol in the Fe-CPD catalytic system. Interestingly, this weak interaction is not observed when the -TMS group in **1a** is replaced with a -$^{t}$Bu group, aligning well with experimental observation (up to 75:25 er for tert-butyl phenyl ketone, see Supplementary Information). This difference may be attributed to the smaller atomic diameter of carbon compared to silicon (Supplementary Fig. 12).

### Asymmetric transfer hydrogenation

Inspired by the deuterium experiments (Fig. 6a), we became interested in adapting this catalytic system for a more user-friendly transfer asymmetric hydrogenation process. Initially, the original **Fe9** (2 mol %) exhibited low catalytic activity for the transfer hydrogenation of **1a** using $^{i}$PrOH as the hydrogen donor, achieving a yield of 30% of **2a** in 12 h. Funk et al. demonstrated that replacing one of the three CO ligands with a weakly coordinating nitrile ligand could significantly enhance catalyst performance[58]. Consequently, we prepared the **Fe5-PhCN** and **Fe9-PhCN** complexes to test their efficacy in asymmetric transfer hydrogenation. To our delight, the PhCN-ligated complexes exhibited excellent catalytic activity at a slightly elevated temperature (70 °C) in a mixed solvent of $^{i}$PrOH/$H_2O$ (15:1) (Fig. 7). The enantiomeric ratios (e.r.) were very similar to those observed in $H_2$-promoted hydrogenation, with only a slight decrease due to the higher

temperature, indicating the formation of the same Fe−H intermediate and hydrogenation transition state. The successful implementation of the asymmetric transfer hydrogenation process further enhances the practicality of this catalytic system, particularly in laboratory settings.

In conclusion, we have successfully developed a new chiral Fe-cyclopentadienone catalyst. This new Fe-complex is characterized by two side arms oriented downward toward the cyclopentadienone plane, and these arms are able to create a steric difference in front of the catalytically active Fe-center. This structural feature was thoroughly explored and explained through experimental and DFT studies. This type of Fe-complex demonstrated excellent catalytic reactivity and selectivity in the asymmetric hydrogenation of acylsilanes, whether aryl- or alkyl-substituted. The broad substrate scope and successful gram-scale synthesis underscore the practicality of this catalytic system. Overall, this study represents a significant breakthrough in the development of chiral Fe-cyclopentadienone catalysts, which may be further applied in various bifunctional asymmetric catalytic transformations in the future.

## Methods

### General procedure for catalytic hydrogenation using $H_2$

Under a nitrogen atmosphere, a 5-mL glass vial equipped with a magnetic stir bar was charged with **Fe9** (9.7 mg, 2 mol %), solvent (1.2 mL $^{i}$PrOH and 0.3 mL $H_2O$), $Me_3NO$ (1.5 mg, 0.02 mmol, 4 mol %), and substrate **1** (0.5 mmol, 1.0 equiv.). The vial was then transferred to a 50-mL autoclave, which was purged with $H_2$ twice (charging with 10 atm $H_2$ and slowly releasing the $H_2$ each time). The autoclave was subsequently charged with $H_2$ to a pressure of 30 atm. The autoclave was stirred and heated in an oil bath at 60 °C for 12 h. After cooling to ambient temperature, the $H_2$ was carefully released. The solvent was

then removed under reduced pressure, and the residue was purified by silica gel column chromatography to obtain the desired product **2**.

## General procedure for catalytic transfer hydrogenation using *i*PrOH

Under a nitrogen atmosphere, a 4-mL glass vial with a screw cap and a magnetic stir bar was charged with **Fe9-PhCN** (6.3 mg, 2 mol %), solvent (1.5 mL *i*PrOH and 0.1 mL $H_2O$), and substrate **1** (0.3 mmol, 1.0 equiv.). The vial was tightly sealed with the screw cap. The mixture was stirred and heated in an oil bath at 70 °C for 12 h. After cooling to room temperature, the solvent was removed under reduced pressure, and the residue was purified by silica gel column chromatography to obtain the desired product **2**.

## Data availability

All data generated and analyzed during this study are included in this Article and its Supplementary Information/Source Data file. Crystallographic data of **Fe4** have been deposited at the Cambridge Crystallographic Data Center, under deposition number CCDC 2446024. Copies of the data can be obtained free of charge via https://www.ccdc.cam.ac.uk/structures/. Data supporting the findings of this manuscript are also available from the corresponding author upon request.

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

## Acknowledgments

We thank the Hong Kong Research Grants Council (21304324, X.T.), start-up fund from the City University of Hong Kong (Project no 9610667, X.T.), the Guangdong Basic and Applied Basic Research Foundation (2024A1515010323, 2025A1515011907, S.-F.N.), and the open research fund of Songshan Lake Materials Laboratory (2023SLABFN16, S.-F.N.) for financial support. We also thank Dr. Ken Shek Man Yiu in the Chemistry department for assistance in structure determination by X-ray crystallography.

## Author contributions

C.X. performed the experiments and collected the data. B.-X.Y. and S.-F.N. performed the DFT calculations. K.-C.L. provided help for the dimensional NMR test. X.Z. provided suggestions and discussions for the project. C.X. and B.-X.Y. contributed equally to this work. X.T. conceived and directed the project and wrote the paper. All the authors discussed the results and commented on the manuscript.

## Competing interests

X.T. and C.X. are inventors of a patent (U.S. Non-Provisional Utility Patent Application No. 19/370,723). X.T. and C.X. declare no other competing interests. All the other authors have no competing interests to declare.
