## [Transparent Peer Review file · Nature Communications]

Rationally Designed Fe-Cyclopentadienone with Unique Orientations for Efficient Asymmetric Hydrogenation of Acylsilanes

Corresponding Author: Professor Xuefeng Tan

Version 0:

Reviewer comments:

Reviewer #1

(Remarks to the Author)

Tan and co-workers developed a new chiral Fe-cyclopentadienone catalyst for asymmetric hydrogenation and hydrogen transfer of acylsilanes, achieving high yields and good enantioselectivity. The catalyst features easily accessible starting materials, facile synthesis, and air-stable products. Crucially, the Fe-complex forms as a single diastereomer, simplifying isolation. Its unique structure incorporates bulky downward-oriented side arms that create a confined chiral environment, enabling broad substrate scope (aryl-, alkenyl-, alkyl-acylsilanes) and gram-scale syntheses with low catalyst loading. It is noteworthy that both asymmetric hydrogenation and hydrogen transfer exhibit comparable enantioselectivity, suggesting the possible involvement of shared Fe-H intermediates. Mechanistic studies reveal the catalyst's structural stability and rigidity during catalysis through experimental and DFT analyses. DFT calculations further elucidate that weak substrate-catalyst interactions in key transition states govern enantiocontrol. The manuscript is well-structured with comprehensive Supporting Information. While this work represents a valuable contribution to asymmetric catalysis, minor concerns remain regarding condition optimization details and DFT methodology.

Comments:

- (1) In the substrate scope investigations, different *i*PrOH/H₂O ratios were employed for the two reaction types (4:1 for asymmetric hydrogenation versus 15:1 for transfer hydrogenation). Why choose these ratios? What is the functional role of H₂O in the reaction system?
- (2) The gram-scale reactions showed complete conversion at S/C=1000 and 300, yet afforded isolated yields of just 87% and 86%. Why? Were any byproducts detected?
- (3) In DFT calculations, the authors employed Fe²⁺ as the catalyst model. Why was Fe⁷⁺, which exhibits optimal catalytic performance, not adopted for these computations? The reasons should be mentioned.
- (4) In line 249, after 'Funk et al. demonstrated that replacing one of the three CO ligands with a weakly coordinating nitrile ligand could significantly enhance catalyst performance', the corresponding literature should be cited.
- (5) The authors mention that replacing one of the three CO ligands with a weakly coordinating nitrile ligand could significantly enhance catalyst performance in asymmetric transfer hydrogenation. Could higher turnover numbers be achieved by employing Fe⁷⁺-PhCN as the catalyst in the asymmetric hydrogenation?
- (6) Relevant reviews and representative research papers on weak interactions in asymmetric hydrogenation should be cited.
- (7) Please check for minor issues in the article, such as the sentence in line 56 and the abbreviation "TMAO."

Reviewer #2

(Remarks to the Author)

This contribution introduces a new chiral symmetric (C₂) modification of a cyclopentadienone ligand for application in Fe(cyclopentadienone)-catalysed hydrogenation and transfer hydrogenation of acylsilanes. The ligand can be rather easily assembled from commercially available L-proline and is obtained as a single enantiomer. A sufficiently wide scope is presented, with generally high enantiomeric ratios. The reason for the enantioselectivity has been traced to weak interactions established with the prochiral substrate, with support from DFT calculations.

I think this paper is potentially impactful, but I would wish to have additional insight. The investigation is restricted to silyl-

substituted ketones, a rather poorly investigated class of substrates in hydrogenation, relative to the more common diaryl-, alkyl-aryl-, and dialkyl ketones. In the abstract, the authors claim that a key factor is the weak interaction between the chiral catalyst and the silyl group, but this interaction involves in effect a C-H bond of one of the silyl methyl substituents. Thus, an analogous interaction could in principle be established with simpler aryl-alkyl ketones. I would with the authors to include in their paper at least one example of a hydrogenation of a representative aryl-alkyl ketone with analogous structure (for instance, phenyl-tert-butyl ketone, or perhaps propiophenone) from both the experimental and DFT point of view. If the key weak interaction is still present, the scope of the study would widen in a very interesting way. If, on the other hand, the weak interaction disappears and the results are poorer, the authors can gain additional insights on the key role of the silyl group.

Additional points:

1. The authors quickly discard the possible intervention of other spin states on the basis of results obtained for the saturated tricarbonyl pre-catalyst. This is not representative. They should rather do the same test on the less saturated (16-electron) dicarbonyl analogues (Int1), or on the dihydrogen adduct of this (Int2). I believe that the system would still remain more stable in the singlet state, but the gap will certainly be must smaller.
2. Line 95: who are catalysts A-D?
3. Line 109: there should more details (or a literature reference) on the Bestmann-Ohira reagent.
4. Line 113: the authors should add the global yield of the catalysts Fe1-10 from L-proline.

Reviewer #3

(Remarks to the Author)

The development of the efficient chiral Fe-cyclopentadienone catalyst for asymmetric catalysis represents a challenging task. In the manuscript, the authors developed a series of novel chiral Fe-cyclopentadienone catalysts through a rational design by simultaneously positioning steric side arms downward to the cyclopentadienone plane and creating two distinct steric environments in front of the catalytically active iron center. The resulting catalysts displayed excellent performance in the asymmetric hydrogenation or transfer hydrogenation of acylsilanes, achieving up to a 96:4 e.r. and a turnover number (TON) of 870. A detailed mechanistic understanding and enantiocontrol insight of the reaction were achieved through experimental and DFT studies. This study represents an efficient strategy for the construction of chiral Fe-cyclopentadienone catalyst and should be helpful to guide the new ligand design. The manuscript and the supporting information are well presented. The manuscript is therefore recommended for the publication in Nat. Commun. after the following points are addressed,

1. As a key point in the design of new Fe-CPD catalysts, the introduction of the ArSO₂ group should be crucial to the success of the present strategy as showed in Figure 1c. However, no data supported this assumption in the manuscript. Some comparative experiments should be provided to further elucidate the necessity of ArSO₂ group in the ligand, including the use of parent chiral Fe-CPD catalyst without a ArSO₂ group and the use of chiral Fe-CPD catalyst with acyl group such as Boc or acetyl instead of a ArSO₂ group for the hydrogenation.
2. The authors screened the substituent at the phenyl ring of ArSO₂ group to establish the optimal catalyst in Figure 2. However, the result in Figure 2 did not display a clear relationship between the substituent at the phenyl ring and the catalytic performance. it seems that the substituent at the phenyl ring is not so critical to catalytic performance. Strangely, the authors did not investigate the Fe-CPD catalyst with the simplest PhSO₂ group as a comparison. Thus, more extensive evaluation on representative sulfonyl groups such as PhSO₂, tBuPhSO₂, MeSO₂ and tBuSO₂ should be performed to clarify the importance of the substituent on the sulfonyl group.
3. keep "Me₃NO" and "TMAO" in Fig. 3 and 4 consistent with those in the footnote in Fig. 3 and 4.

Version 1:

Reviewer comments:

Reviewer #1

(Remarks to the Author)

Having addressed all the issues raised during the review process, the manuscript in its current form satisfies the requirements for publication in this journal.

Reviewer #2

(Remarks to the Author)

The revision is satisfactory in terms of the additional work carried out to address my criticism, but I think that this additional work is not sufficiently well highlighted in the revised manuscript.

1. I appreciate that the authors have carried out my suggested additional experiment with phenyl-tert-butyl ketone and related calculations. The results are very interesting and would highly deserve to be featured, or at least mentioned, in the main text, whereas currently that have only been added to the SI.
2. In my opinion, the description of the catalytic mechanism (Figure S2 and the associated text) deserves to be featured in the main text, rather than relegated to the Supporting Information. At least the nature and spin state of the catalyst resting

state (³Int2) should be highlighted in the main text.

3. The revised description of the DFT gives the impression that the MECPs were located and optimized. However, the SI does not report the geometries and energies of these critical points. If they have not been located, they should not be explicitly shown in Figure S2. I would recommend, however, to also include the optimized MECPs, particularly MECP2, because this is placed between the rate-determining intermediate (³Int2) and the rate-determining transition state (¹TS2). Thus, the energy span of the cycle may actually be greater than 27 kcal/mol, if the MECP2 is located at higher energy than TS2.

4. There is no need for MECP3, because it is the same as MECP1.

5. All species in Figure S2 should be redrawn to scale.

Reviewer #3

(Remarks to the Author)

The authors have well addressed the reviewers' concerns, this reviewer therefore recommend the manuscript for the publication as it is.

Response to the reviewers' comments

Reviewer #1

General comments: Tan and co-workers developed a new chiral Fe-cyclopentadienone catalyst for asymmetric hydrogenation and hydrogen transfer of acylsilanes, achieving high yields and good enantioselectivity. The catalyst features easily accessible starting materials, facile synthesis, and air-stable products. Crucially, the Fe-complex forms as a single diastereomer, simplifying isolation. Its unique structure incorporates bulky downward-oriented side arms that create a confined chiral environment, enabling broad substrate scope (aryl-, alkenyl-, alkyl-acylsilanes) and gram-scale syntheses with low catalyst loading. It is noteworthy that both asymmetric hydrogenation and hydrogen transfer exhibit comparable enantioselectivity, suggesting the possible involvement of shared Fe-H intermediates. Mechanistic studies reveal the catalyst's structural stability and rigidity during catalysis through experimental and DFT analyses. DFT calculations further elucidate that weak substrate-catalyst interactions in key transition states govern enantiocontrol. The manuscript is well-structured with comprehensive Supplementary Information.

Our response: We thank the referee for the positive comments.

Remarks: While this work represents a valuable contribution to asymmetric catalysis, minor concerns remain regarding condition optimization details and DFT methodology.

Remark 1: In the substrate scope investigations, different *i*PrOH/H₂O ratios were employed for the two reaction types (4:1 for asymmetric hydrogenation versus 15:1 for transfer hydrogenation). Why choose these ratios? What is the functional role of H₂O in the reaction system?

Our response: Thank you for this insightful comment. Indeed, H₂O plays a crucial role in our catalytic system. We conducted the following three catalytic reactions under 60 °C, and the results demonstrate that the inclusion of H₂O significantly enhances catalytic reactivity. This enhancement may be attributed to H₂O acting as a proton shuttle, facilitating the formation of a bifunctional transition state. We have included experiments 1 and 2 in the Supplementary Information, detailed in Table S3, entries 7-8.

1. **toluene** as solvent, **2a**, 50%, 92 : 8 er
2. ***i*PrOH** as solvent, **2a**, 35%, 93 : 7 er
3. ***i*PrOH/H₂O** (4:1), **2a**, 93%, 93 : 7 er

Regarding the different *i*PrOH/H₂O ratios used in hydrogenation (4:1) and transfer hydrogenation (15:1), we performed several reactions for comparison. By keeping the total solvent volume fixed at 2 mL, we found that a higher concentration of *i*PrOH improves

reactivity. This result is imaginable as the transfer hydrogenation using *i*PrOH as the hydrogen source. Although the results for the 9:1 ratio were comparable to those for the 15:1 ratio, we opted for the 15:1 ratio in subsequent studies to ensure broader substrate scope and generality.

Remark 2: The gram-scale reactions showed complete conversion at S/C=1000 and 300, yet afforded isolated yields of just 87% and 86%. Why? Were any byproducts detected?

Our response: Yes, in the high TON test experiment, we indeed observed a side product, benzaldehyde, particularly in cases of low conversions. This side reaction has been documented in the literature (J. Chem. Res. 2000, 2000, 404-405. DOI: 10.3184/030823400103167868), as indicated by the following transition state. When the catalytic reaction is conducted at high catalyst loading (2 mol%), the slow side reaction is not significant. However, at low catalyst loading (0.1 mol%), this side reaction becomes negligible. To clarify this issue, we have added the following sentence to the gram-scale synthesis part: “The low yield is due to the slow decomposition of **2a** in water, which produces benzaldehyde.”

Remark 3: In DFT calculations, the authors employed **Fe4** as the catalyst model. Why was **Fe9**, which exhibits optimal catalytic performance, not adopted for these computations? The reasons should be mentioned.

Our response: We appreciate the referee's concern regarding the use of **Fe9**, the best-performing catalyst, as a DFT model for a more accurate representation of the system. However, we were unable to obtain a crystal structure for **Fe9**. As noted in the manuscript, the catalyst structure is quite unique and unexpected, prompting us to focus on illustrating the catalyst structure more clearly. Therefore, we chose to use a well-established crystal structure as the model for our calculations, allowing for a straightforward comparison between the calculated results and the actual structure. To clarify this point, we have revised the first sentence in the DFT section to read: “Next, experimental and DFT studies were conducted using **Fe4**, which has been crystallographically characterized, as the standard model to elucidate the catalytic mechanism and enantiocontrol insight (Fig. 6).”

Remark 4: In line 249, after ‘Funk et al. demonstrated that replacing one of the three CO

ligands with a weakly coordinating nitrile ligand could significantly enhance catalyst performance', the corresponding literature should be cited.

Our response: We appreciate the referee's suggestion. Due to an oversight on our part, this reference was cited as reference 55 in our original manuscript but was not annotated in the main text. We have now added this citation as reference 58.

Remark 5: The authors mention that replacing one of the three CO ligands with a weakly coordinating nitrile ligand could significantly enhance catalyst performance in asymmetric transfer hydrogenation. Could higher turnover numbers be achieved by employing **Fe9-PhCN** as the catalyst in the asymmetric hydrogenation?

Our response: We appreciate the referee's constructive suggestion. By employing **Fe9-PhCN** as the catalyst, we conducted catalytic hydrogenation at S/C = 1000. However, analysis of the reaction revealed comparable TON values of **Fe9-PhCN** (810) and **Fe9** (870).

Remark 6: Relevant reviews and representative research papers on weak interactions in asymmetric hydrogenation should be cited.

Our response: We appreciate the referee's friendly reminder. Due to the limitation on the number of references, we have added only two citations: references 56 and 57. Reference 56 describes a very similar C–H···O interaction, while reference 57 provides an excellent review of weak interactions in asymmetric hydrogenation.

Remark 7: Please check for minor issues in the article, such as the sentence in line 56 and the abbreviation "TMAO."

Our response: We appreciate the referee's kind reminder. We have changed all TMAO to Me₃NO.

Reviewer #2

General comment: This contribution introduces a new chiral symmetric (C₂) modification of a cyclopentadienone ligand for application in Fe(cyclopentadienone)-catalysed hydrogenation and transfer hydrogenation of acylsilanes. The ligand can be rather easily assembled from commercially available *L*-proline and is obtained as a single enantiomer. A sufficiently wide scope is presented, with generally high enantiomeric ratios. The reason for the enantioselectivity has been traced to weak interactions established with the prochiral substrate, with support from DFT calculations.

Our response: We thank the referee for the positive comments.

Remark 1: I think this paper is potentially impactful, but I would wish to have additional insight. The investigation is restricted to silyl-substituted ketones, a rather poorly investigated class of substrates in hydrogenation, relative to the more common diaryl-, alkyl-aryl-, and dialkyl ketones. In the abstract, the authors claim that a key factor is the weak interaction between the chiral catalyst and the silyl group, but this interaction involves in effect a C-H bond of one of the silyl methyl substituents. Thus, an analogous interaction could in principle be established with simpler aryl-alkyl ketones. I would wish the authors to include in their paper at least one example of a hydrogenation of a representative aryl-alkyl ketone with analogous structure (for instance, phenyl-*tert*-butyl ketone, or perhaps propiophenone) from both the experimental and DFT point of view. If the key weak interaction is still present, the scope of the study would widen in a very interesting way. If, on the other hand, the weak interaction disappears and the results are poorer, the authors can gain additional insights on the key role of the silyl group.

Our response: We sincerely thank the reviewer for the valuable comment. In response, we have performed additional experiments by replacing the silyl substituent with a *tert*-butyl group. The silyl-substituted substrate exhibits a higher er than the *tert*-butyl-substituted one. Results are listed as follow:

Theoretical calculations indicate that this phenomenon mainly arises from the larger atomic radius of silicon compared to carbon, which enables the silyl group to form a stronger C–H···O weak interaction with the catalyst ($b_6 = 2.354 \text{ \AA}$). Although the *tert*-butyl group can also participate in a similar interaction, the longer interaction distance ($b_9 = 2.614 \text{ \AA}$) leads to a weaker stabilizing effect relative to the silyl group. This comparison has been added in the Supplementary Information (Figure S5).

Fig. S5 | Non-covalent interaction analysis of transition state TS_{S-t-Bu} and TS_{R-t-Bu} .

Remark 2: The authors quickly discard the possible intervention of other spin states on the basis of results obtained for the saturated tricarbonyl pre-catalyst. This is not representative. They should rather do the same test on the less saturated (16-electron) dicarbonyl analogues (Int1), or on the dihydrogen adduct of this (Int2). I believe that the system would still remain more stable in the singlet state, but the gap will certainly be must smaller.

Our response: We sincerely appreciate the reviewer's insightful suggestion. Accordingly, we have carried out theoretical calculations for all Fe-containing complexes involved in the catalytic cycle, considering different spin states (see Figure S1). As correctly pointed out by the reviewer, the Fe center indeed exhibits distinct spin multiplicities depending on its electronic configuration. However, the calculated Gibbs free energy differences between the singlet and triplet states of **Int1**, **Int2**, and **Int-H** are relatively small. Based on these results, we have revised the proposed reaction pathway (see Figure S2). Specifically, $^1\text{Fe}^{\text{IV}}_{\text{cal}}$ initiates from the singlet state, followed by CO dissociation and a spin crossover via the minimum energy crossing point (**MECP1**) to afford the more stable triplet intermediate $^3\text{Int1}$.

Subsequent association with H₂ yields the triplet intermediate **³Int2**, which then undergoes a spin inversion through **MECP2** to produce the singlet intermediate **¹Int2**. The reaction proceeds via **¹TS2** along the singlet potential energy surface to furnish the final product. We also attempted to locate the triplet **TS2**, but unfortunately, this transition state has not yet been identified. The corresponding description in the Supporting Information has been modified accordingly to read as: “As illustrated in Figure S2, the catalytic cycle starts from activation of the pre-catalyst **¹Fe4_{cal}** initiates with Me₃NO-assisted CO ligand dissociation, accompanied by the release of CO₂ and Me₃N. This process yields intermediate **¹Int1** with a favorable free energy of -51.3 kcal·mol⁻¹. The corresponding transition state **¹TS1** presents an activation barrier of 12.2 kcal·mol⁻¹, indicating both kinetic accessibility and thermodynamic favorability. Subsequently, **¹Int1** undergoes a spin crossover via the minimum energy crossing point (**MECP1**), affording the more stable triplet intermediate **³Int1**. Association of H₂ to **³Int1** generates intermediate **³Int2**, which similarly undergoes a spin crossover through **MECP2**, leading to the singlet intermediate **¹Int2**. The latter proceeds via **¹TS2**, involving H–H bond cleavage ($\Delta G^\ddagger = 27.0$ kcal·mol⁻¹), representing the rate-determining step. **Int-H** then reacts with **1a** to form **¹Int3**, followed by a concerted hydrogen transfer through **¹TS_S** ($\Delta G^\ddagger = 13.7$ kcal·mol⁻¹) to afford the desired product **2a**. This final step is thermodynamically favorable, and **¹Int1** undergoes spin crossover via **MECP3** to regenerate **³Int1**, thereby completing the catalytic cycle.”

Fig. S1 | The relative free energies of Fe complexes in different spin states.

Fig. S2 | DFT calculated free energies of the whole catalytic cycles.

Remark 3: Line 95: who are catalysts A-D?

Our response: We apologize for the confusion caused by our oversight. Catalysts A-D refer to the chiral Fe-cyclopentadienone complexes described in references 44 and 48-51. We initially included these structures in Figure 1 but ultimately decided to remove them for clarity. Therefore, we have revised the sentence to read: "This conclusion is supported by the crystal structures of the catalysts described in earlier reports (44, 48-51)."

Remark 4: Line 109: there should more details (or a literature reference) on the Bestmann-Ohira reagent.

Our response: We appreciate the referee's kind reminder. We have now provided the full name of the Bestmann-Ohira reagent, "dimethyl (1-diazo-2-oxopropyl)phosphonate," in both the text and the caption of Figure 2.

Remark 5: Line 113: the authors should add the global yield of the catalysts **Fe1-10** from L-proline.

Our response: We appreciate this valuable suggestion, as providing global yields can offer readers a clear perspective on the synthesis. Since all complexes share the common synthon P3, which is a known compound, we have detailed the yield of P3 from L-Proline as 63.5% in the Supplementary Information. Additionally, the global yields of the Fe-complexes are indicated in the manuscript, stating: "Starting from the common synthon P3, the final complexes **Fe1-12** achieved a global yield ranging from 12% to 46% (see Supplementary Information)."

Reviewer #3

General comment: The development of the efficient chiral Fe-cyclopentadienone catalyst for asymmetric catalysis represents a challenging task. In the manuscript, the authors developed a series of novel chiral Fe-cyclopentadienone catalysts through a rational design by simultaneously positioning steric side arms downward to the cyclopentadienone plane and creating two distinct steric environments in front of the catalytically active iron center. The resulting catalysts displayed excellent performance in the asymmetric hydrogenation or transfer hydrogenation of acylsilanes, achieving up to a 96:4 e.r. and a turnover number (TON) of 870. A detailed mechanistic understanding and enantiocontrol insight of the reaction were achieved through experimental and DFT studies. This study represents an efficient strategy for the construction of chiral Fe-cyclopentadienone catalyst and should be helpful to guide the new ligand design. The manuscript and the Supplementary Information are well presented.

Our response: We thank the referee for the positive comments.

Remarks: The manuscript is therefore recommended for the publication in Nat. Commun. after the following points are addressed.

Remark 1: As a key point in the design of new Fe-CPD catalysts, the introduction of the ArSO₂ group should be crucial to the success of the present strategy as showed in Figure 1c. However, no data supported this assumption in the manuscript. Some comparative experiments should be provided to further elucidate the necessity of ArSO₂ group in the ligand, including the use of parent chiral Fe-CPD catalyst without a ArSO₂ group and the use of chiral Fe-CPD catalyst with acyl group such as Boc or acetyl instead of a ArSO₂ group for the hydrogenation.

Our response: We appreciate the referee's insightful comment. In response to the referee's suggestion, we synthesized the Boc-substituted and bare H-substituted complexes, **Fe13** and **Fe14**. The results indicate that the Boc-substituted catalyst **Fe13** is slightly inferior in enantio-induction compared to the -SO₂Ar substituted catalyst, but it exhibits comparable catalytic reactivity. However, **Fe14** demonstrates significantly lower catalytic reactivity and enantiocontrol ability. We have included the synthesis of **Fe13** and **Fe14** in the Supplementary Information, and the catalytic results are presented in Table S1.

Remark 2: The authors screened the substituent at the phenyl ring of ArSO₂ group to establish the optimal catalyst in Figure 2. However, the result in Figure 2 did not display a clear relationship between the substituent at the phenyl ring and the catalytic performance. It seems that the substituent at the phenyl ring is not so critical to catalytic performance. Strangely, the authors did not investigate the Fe-CPD catalyst with the simplest PhSO₂ group as a comparison. Thus, more extensive evaluation on representative sulfonyl groups such as PhSO₂, *t*BuPhSO₂, MeSO₂ and *t*BuSO₂ should be performed to clarify the importance of the substituent on the sulfonyl group.

Our response: Thank you for your constructive suggestions, which may help us gain a better understanding of the catalytic model. We synthesized two representative catalysts, **Fe1** and **Fe2**, with EtSO₂- and PhSO₂- substituents. When applied to the catalytic reaction, these two catalysts demonstrated comparable catalytic reactivity and enantiocontrol ability to other RSO₂- substituents. These comparisons suggest that a substituent (e.g., RSO₂-, Boc-) on the nitrogen enhances both reactivity and enantioselectivity, with RSO₂- performing slightly better than the Boc substituent. However, the R group in RSO₂- shows only a slight difference in enantioselectivity, which is not very significant. To reflect this information in the manuscript, we have added **Fe1** and **Fe2** to Figure 2. And we also added a sentence in the manuscript: “Moreover, the different types of R substituents on the sulfonyl group does not significantly influence either catalytic reactivity or enantioselectivity.”

Remark 3: keep “Me₃NO” and “TMAO” in Fig. 3 and 4 consistent with those in the footnote in Fig. 3 and 4.

Our response: We appreciate the referee’s kind reminder. We have changed all TMAO to Me₃NO.

Response to the reviewers' comments (the 2nd round)

Reviewer #1

General comments: Having addressed all the issues raised during the review process, the manuscript in its current form satisfies the requirements for publication in this journal.

Our response: We appreciate the referee's efforts in reviewing our paper and are grateful for the acceptance of our manuscript.

Reviewer #2

General comment: The revision is satisfactory in terms of the additional work carried out to address my criticism, but I think that this additional work is not sufficiently well highlighted in the revised manuscript.

Our response: We thank the referee's suggestion and the following actions have been adopted.

Remark 1: I appreciate that the authors have carried out my suggested additional experiment with phenyl-tert-butyl ketone and related calculations. The results are very interesting and would highly deserve to be featured, or at least mentioned, in the main text, whereas currently that have only been added to the SI.

Our response: This result has now been incorporated into the main text, following the non-covalent interaction (NCI) analysis, as follows: "Interestingly, this weak interaction is not observed when the -TMS group in 1a is replaced with a -'Bu group, which aligns well with experimental observations (up to 75:25 er for tert-butyl phenyl ketone; see Supplementary Information). This difference may be attributed to the smaller atomic diameter of carbon compared to silicon (Supplementary Fig. 5)."

Remark 2: In my opinion, the description of the catalytic mechanism (Figure S2 and the associated text) deserves to be featured in the main text, rather than relegated to the Supporting Information. At least the nature and spin state of the catalyst resting state (³Int2) should be highlighted in the main text.

Our response: Thank you for your suggestion. We have now incorporated the calculated reaction pathway into Figure 6c, as well as the spin states are indicated and discussed in the main text.

Remark 3: The revised description of the DFT gives the impression that the MECPs were located and optimized. However, the SI does not report the geometries and energies of these critical points. If they have not been located, they should not be explicitly shown in Figure S2. I would recommend, however, to also include the optimized MECPs, particularly MECP2,

because this is placed between the rate-determining intermediate (³Int2) and the rate-determining transition state (¹TS2). Thus, the energy span of the cycle may actually be greater than 27 kcal/mol, if the MEC2 is located at higher energy than TS2.

Our response: Thank you for your reminder, we have now added the coordinates of MEC2 in the Supplementary Information. In the meanwhile, the energy level of MEC2 has been indicated as -39.4 kcal mol⁻¹ in the corrected Figure S2 and Figure 6c.

Remark 4: There is no need for MEC3, because it is the same as MEC1.

Our response: We have deleted both MEC1 and MEC3 in the corrected Figure S2.

Remark 5: All species in Figure S2 should be redrawn to scale.

Our response: Thank you for your kind reminder. Figure S2 has been redrawn to scale.

Reviewer #3

General comment: The authors have well addressed the reviewers' concerns, this reviewer therefore recommend the manuscript for the publication as it is.

Our response: We appreciate the referee's efforts in reviewing our paper and are grateful for the acceptance of our manuscript.